# Effect of Interferon Gamma on Ebola Virus Infection of Primary Kupffer Cells and a Kupffer Cell Line

**DOI:** 10.3390/v15102077

**Published:** 2023-10-11

**Authors:** José A. Aguilar-Briseño, Jonah M. Elliff, Justin J. Patten, Lindsay R. Wilson, Robert A. Davey, Adam L. Bailey, Wendy J. Maury

**Affiliations:** 1Department of Microbiology and Immunology, University of Iowa, Iowa City, IA 52242, USA; jose-aguilarbriseno@uiowa.edu; 2Graduate Program in Immunology, University of Iowa, Iowa City, IA 52242, USA; jonah-elliff@uiowa.edu; 3Department of Virology, Immunology, and Microbiology, National Emerging Infectious Diseases Laboratories, Boston University, Boston, MA 02118, USA; jjpatten@bu.edu (J.J.P.); radavey@bu.edu (R.A.D.); 4Department of Pathology and Laboratory Medicine, University of Wisconsin School of Medicine and Public Health, Madison, WI 53726, USA; liw145@pitt.edu (L.R.W.); albailey@wisc.edu (A.L.B.)

**Keywords:** macrophage, Kupffer cell, interferon gamma, Ebola virus, filovirus

## Abstract

Ebola virus disease (EVD) represents a global health threat. The etiological agents of EVD are six species of Orthoebolaviruses, with *Orthoebolavirus zairense* (EBOV) having the greatest public health and medical significance. EVD pathogenesis occurs as a result of broad cellular tropism of the virus, robust viral replication and a potent and dysregulated production of cytokines. In vivo, tissue macrophages are some of the earliest cells infected and contribute significantly to virus load and cytokine production. While EBOV is known to infect macrophages and to generate high titer virus in the liver, EBOV infection of liver macrophages, Kupffer cells, has not previously been examined in tissue culture or experimentally manipulated in vivo. Here, we employed primary murine Kupffer cells (KC) and an immortalized murine Kupffer cell line (ImKC) to assess EBOV-eGFP replication in liver macrophages. KCs and ImKCs were highly permissive for EBOV infection and IFN-γ polarization of these cells suppressed their permissiveness to infection. The kinetics of IFN-γ-elicited antiviral responses were examined using a biologically contained model of EBOV infection termed EBOV ΔVP30. The antiviral activity of IFN-γ was transient, but a modest ~3-fold reduction of infection persisted for as long as 6 days post-treatment. To assess the interferon-stimulated gene products (ISGs) responsible for protection, the efficacy of secreted ISGs induced by IFN-γ was evaluated and secreted ISGs failed to block EBOV ΔVP30. Our studies define new cellular tools for the study of EBOV infection that can potentially aid the development of new antiviral therapies. Furthermore, our data underscore the importance of macrophages in EVD pathogenesis and those IFN-γ-elicited ISGs that help to control EBOV infection.

## 1. Introduction

Filoviruses are important viral pathogens that represent a serious global health concern. The family *Filoviridae* belongs to the order *Mononegavirales* and the genus *Orthoebolavirus* is composed of six viral species: *Orthoebolavirus zairense*, *Orthoebolavirus sudanense*, *Orthoebolavirus bundibugyoense*, *Orthoebolavirus taiense*, *Orthoebolavirus restonense* and *Orthoebolavirus bombaliense* [1,2,3]. Of these, Ebola virus (EBOV), representing the species *Orthoebolavirus zairense*, has the greatest public health and medical significance [4]. Orthoebolaviruses are enveloped, pleomorphic viruses that contain a negative-sense single-stranded RNA genome of ~19 kb. Infection with EBOV induces a wide range of clinical manifestations encompassing fever, rash, gastrointestinal distress, malaise and myalgia. Patients who subsequently develop fatal disease can manifest hemorrhagic fever, hypovolemic shock and/or organ failure with a mortality rate of up to 90% [4,5]. In 2019, the FDA approved the first vaccine for the prevention of Ebola virus disease (EVD) which consists of recombinant vesicular stomatitis virus (rVSV) that expresses the EBOV glycoprotein (GP). This vaccine confers substantial protection against EVD [6]; however, it provides little to no cross-protection against other ebolaviruses in animal models [7,8,9]. A pan-filovirus vaccine is needed, and such vaccines are currently under development [10,11,12,13]. 

Tissue mononuclear phagocytes, e.g., macrophages and dendritic cells (DCs), are thought to be the first cells in the body infected [14,15,16]. These cells both respond to and elicit innate immune responses that, depending on the situation, ameliorate or exacerbate the associated disease [17,18,19]. Peritoneal macrophages polarized with interferon gamma (IFN-γ) (M1 polarization) stimulates the production of a large group of interferon-stimulated genes (ISGs), suppressing viral replication in this cell population and protecting mice from EBOV disease [18]. However, M1 polarization of tissue macrophages can be a double-edged sword, as the production of proinflammatory soluble factors at late stages of EBOV infection is associated with worse outcomes [20,21]. In contrast, IL-4/IL-13 treatment of peritoneal macrophages that induces M2a polarization enhances virus infection of the cells early on and sustains them as viral targets via upregulation of C-type lectins on the cell surface [17]. Hence, the microenvironment of tissue mononuclear phagocytes affects both the ability of these cells to support EBOV infection and the cytokines produced. While macrophage infection affects both the control of EBOV replication and the immunopathogenesis associated with infection, details of the role of these cells during infection remain incompletely understood. In part, this is due to the limited availability of cell lines that are easy to work with and accurately recapitulate various aspects of tissue macrophages.

Tissue phagocytes also serve as vehicles for EBOV spread. Infected phagocytes (i.e., DCs) travel to the regional lymph nodes where viral replication occurs followed by viremia and viral dissemination to a variety of organs and tissues [16]. The liver is one such organ that becomes infected early during EBOV infection where the tissue-resident macrophages, Kupffer cells (KCs), support infection as well as drive inflammatory responses, leading to liver damage [14,15,22]. However, the interaction of EBOV with KCs has been poorly explored to date. 

Here, we phenotypically characterize murine KCs and an adherent, easily manipulatable macrophage model line, immortalized mouse Kupffer cells (ImKCs), and found that this line expresses macrophage-specific and, more specifically, Kupffer-cell-specific genes. Further, cytokine-induced polarization-specific markers were comparable between the two cell populations, demonstrating that ImKCs serve as an easily manipulatable proxy for Kupffer cells. Under non-polarized conditions, KCs and ImKCs were highly permissive for EBOV-eGFP, and the use of ImKCs allowed us to study EBOV infection kinetics and quality of the associated macrophage immune response using both authentic EBOV and an EBOV model system. As we previously observed in murine peritoneal macrophages [18], infection was robustly inhibited by IFN-γ pre-treatment of the cells. The duration of IFN-γ-elicited antiviral activity was examined and we found that the profound inhibitory effect of IFN-γ on EBOV infection of ImKCs was transient, with much of the inhibition conferred by IFN-γ waning within a 24 h period. However, a more modest ~three-fold inhibition of virus infection persisted for as long as 6 days following IFN-γ treatment. We also assessed if secreted interferon-stimulated genes (ISGs) contributed to the IFN-γ-induced protection and found that the secretome was not effective at blocking EBOV infection. These data provide insights into the ISGs and the duration of the antiviral effect of IFN-γ and underscore the importance of macrophages in EVD pathogenesis.

## 2. Materials and Methods

### 2.1. Ethics Statement

The study was conducted in strict accordance with the Animal Welfare Act and the recommendations in the Guide for the Care and Use of Laboratory Animals of the National Institutes of Health (University of Iowa (UI) Institutional Assurance Number: #A3021-01). All animal procedures were approved by the UI Institutional Animal Care and Use Committee (IACUC) which oversees the administration of the IACUC protocols, and the study was performed in accordance with the IACUC guidelines (Protocol #1031280).

### 2.2. Primary Kupffer Cell Isolation

Wild-type C57BL/6 mice were a kind gift from Dr. John Harty (University of Iowa). Mice were maintained in agreement with IACUC guidelines at the UI. Primary Kupffer cells were isolated from wild-type C57B/6 mice as previously reported [23]. Briefly, livers were excised, finely chopped, and digested in 10 mL RPMI media containing 1 mg/mL of type IV collagenase (Thermo Fisher Scientific, Waltham, MA, USA, #17104019) for 30 min at 37 °C. Digested tissue was mashed through a 100 μM cell strainer. Hepatocytes were separated from other cells in the liver suspension by low-brake low-speed centrifugation (50× *g*, 3 min, room temperature). Hepatocyte-free suspensions were centrifuged for 7 min, 485× *g*, at room temperature in a 20% Percoll (Sigma-Aldrich, Burlington, MA, USA, #P4937-100ML)/80% HBSS (Gibco/Thermo Fisher Scientific, Waltham, MA, USA, #14025-092) gradient. Supernatants were removed and cell pellets were subject to red blood cell lysis with lab-made lysis buffer (150 mM NH_4_Cl, 10 mM KHCO_3_, 0.1 mM Na_2_EDTA). For macrophage polarization, cells were plated and treated with polarizing cytokines as described below. Cells were plated and sent to the National Emerging Infectious Diseases Laboratories (NEIDL) (Boston, MA, USA). Kupffer cells were treated with polarizing cytokines (see below) and infected with EBOV-eGFP. KCs were phenotypically characterized by flow cytometry by surface staining for macrophage- and KC-associated markers as described below.

### 2.3. Generation of EBOV VP30-Expressing Lentivirus

The coding sequence for EBOV VP30 was extracted from the ZEBOV genome (NCBI Genbank accession ID: NC_002549.1) and codon-optimized for expression in mammalian cells. This nucleic acid was produced as a gBlocks gene fragment (IDT Corporation, Coralville, IA, USA), containing a stop codon and homology arms for Gibson Assembly into the pLV-EF1a-IRES-Hygro vector (Addgene, Watertown, MA, USA, #85134). The resulting pLV-EF1a-VP30-IRES-Hygro construct was confirmed to have the desired insert via Sanger sequencing, then co-transfected into HEK-293T cells with lentiviral packaging plasmids using Lipofectamine 2000 (Thermo Fisher Scientific, Waltham, MA, USA, #11668030). Lentiviral production was confirmed in the supernatant two days later via Lenti-X GoStix (Takara, Kusatsu, Shiga, Japan, #631280), followed by transduction of target cells with filtered HEK-293T supernatant. Target cell transduction was confirmed by subjecting cells to a hygromycin kill curve with mock-transduced cells as a reference. VP30^+^ cells were then subsequently passaged in the presence of hygromycin, as determined by the kill curve.

### 2.4. Cell Lines

ImKCs are commercially available (https://www.sigmaaldrich.com/deepweb/assets/sigmaaldrich/product/documents/371/711/scc119ds.pdf, accessed on 4 October 2023) and were maintained in RPMI supplemented with 10% FBS, penicillin (100 U/mL) and streptomycin (100 μg/mL). Vero cells were grown in DMEM supplemented with 10% FBS, penicillin (100 U/mL) and streptomycin (100 μg/mL). EBOV VP30-expressing Vero cells have been described [24] and were maintained and cultured as regular Vero cells. EBOV VP30-expressing ImKCs were generated by transduction of a VP30-encoding pLV-EF1α-VP30-IRES-hygro_v2.1-hygromycin into ImKCs. Cells were selected with hygromycin (300 μg/mL). VP30-expression of the bulk population was confirmed by Western blot using a rabbit anti-EBOV VP30 polyclonal antisera (IBT Bioservices, Rockville, MD, USA, #0301-048). VP30-expression in ImKCs was further validated by their ability to support EBOV ΔVP30-dependent infection. EBOV VP30-expressing ImKCs were maintained as parental ImKCs. All cell lines used in our experiments tested negative for *Mycoplasma* spp. using a commercially available PCR assay (Bulldog Bio, Portsmouth, NH, USA, #25233).

### 2.5. Primary KC and ImKCs Phenotyping

Primary KCs and ImKCs were immunostained for surface markers. Fixable Viability Dye eFluor 780 (1:1000, #65-0865), CD11b BV421 (1:75, clone M1/70, #404-0112-80) and TIM-4 PerCP eFlour 710 (1:75, clone RMT4-54, #12-5866-82) were purchased from Thermo Fisher (Thermo Fisher Scientific, Waltham, MA, USA). CD45.2 PE (1:5, clone 104, #109808), CD45.2 BV421 (1:75, clone 104, #109831), F4/80 APC (1:75, clone BM8, #123116), CLEC4F AF647 (1:75, clone 3E3F9, #156803), CLEC2 PE (1:75, clone 17D9/CLEC-2, #146103), CD14 PE (1:75, clone Sa-14-2, #123309), TLR4 APC (1:75, clone SA15-21, #145405) were purchased from BioLegend (San Diego, CA, USA). Unstained cells as well as fluorescent minus one (FMO) control samples were used as controls on every staining. All stainings were performed in the presence of the Fc receptor’s blocker monoclonal antibody (Bio X Cell, Lebanon, NH, USA, 20 μg/mL, clone 2.4G2, #BE0307). Samples were measured on a CytoFLEX cytometer (Beckman Coulter, Brea, CA, USA). Data were analyzed using the FlowJo software v10.8.2 (BD Biosciences, Franklin Lakes, NJ, USA).

### 2.6. Viruses

All experiments with the replication-competent EBOV were performed in a NEIDL Biosafety Level 4 (BSL4) laboratory. The recombinant EBOV variant Mayinga expressing enhanced GFP (EBOV-eGFP) was generated and characterized as previously described [25]. EBOV ΔVP30 was derived from the EBOV Mayinga strain and kindly provided by Peter Halfmann (University of Wisconsin). Stocks of EBOV ΔVP30 were propagated and characterized as previously reported [24]. Briefly, the virus was propagated by infecting EBOV VP30-expressing Vero cells at low MOI (~0.005) and collecting supernatants at 5 dpi. The resulting supernatants were filtered through a 45-micron filter and purified by ultra-centrifugation (133,907× *g* maximum, 4 °C, 2 h) through a 20% sucrose cushion. Stocks of EBOV ΔVP30 were resuspended in PBS, stored at −80 °C until used and titered on Vero VP30 cells. Both viruses used in these studies encoded a reporter gene, GFP, that was used to assess virus infection. 

### 2.7. Macrophage Polarization

Polarization of primary KCs and ImKCs was achieved by culturing cells for 24 h in media containing 20 ng/mL IFN-γ (Cell Sciences, Newburyport, MA, USA, #CRI001B) or 20 ng/mL IL-4 (BioLegend, San Diego, CA, USA, #574302) + 20 ng/mL of IL-13 (BioLegend, San Diego, CA, USA, #575902). Following polarization, media were removed and replaced with culturing media without cytokines and harvested for RNA or infected with the virus. Macrophage polarization was validated by qRT-PCR.

### 2.8. In Vitro Infections

For maximum biocontainment laboratory studies, 96-well plates containing KCs (10^5^ cells) and/or ImKCs (10^5^ cells) were mock-treated or treated with IFN-γ (20 ng/μL) 48 h prior to infection with 10^4^ or 10^5^ particles of EBOV-eGFP for 48 h. For EBOV ΔVP30-experiments under BSL2^+^ conditions, 24-well plates containing EBOV VP30-expressing ImKCs (2.5 × 10^5^ cells) were treated with 20 ng/mL of IFN-γ for 24 h followed by infection with an MOI of 1 or 10 of EBOV ΔVP30 for 48 to 60 h as noted in figure legends. For RNA analysis, cells were harvested on TRIzol and stored at 4 °C. To assess infection, infected populations were lifted, washED 1× and resuspended in FACS buffer. GFP expression was measured by a Calibur (BD) or CytoFLEX cytometer (Beckman Coulter, Brea, CA, USA). Data were analyzed using the FlowJo software v10.8.2 (BD Biosciences, Franklin Lakes, NJ, USA).

### 2.9. IFN-γ Protection over Time

For assessing the protection provided by IFN-γ over time, 24-well plates containing EBOV VP30-expressing ImKCs were treated with IFN-γ for 24 h and infected with an MOI of 10 of EBOV ΔVP30 for 60 h starting at 0 h, 24 h, 48 h, 96 h, and 144 h after removing IFN-γ. Analysis of RNA and GFP expression was carried out as described above. To understand the impact of IFN-γ on cell viability throughout the duration of the experiment, a luciferase-based ATPlite™ assay was used (PerkinElmer, Waltham, MA, USA, #A22066). Briefly, 96-well plates containing EBOV VP30-expressing ImKCs (1 × 10^4^ cells) were treated with IFN-γ for 24 h and lysed at 0 h, 24 h, 48 h, 96 h, and 144 h after removing IFN-γ. Luciferase-containing substrate provided by the manufacturer was added directly to cells in the plate, transferred to white-bottomed plates and luminescence was measured by a plate reader (Tecan Infinite 200 Pro, Tecan, Mannedorf, Switzerland) according to the manufacturer’s protocol. 

### 2.10. Focus-Forming Assay

EBOV VP30-expressing Vero cells were seeded at a density of 2.5 × 10^4^ cells per well in flat-bottomed 96-well tissue culture plates. The following day, medium was removed and replaced with 100 μL of 10-fold serial dilutions of ΔVP30-EBOV. Two hours later, 135 μL of methylcellulose overlay was added. Plates were incubated for 3 days and then fixed with 4% paraformaldehyde in phosphate-buffered saline for 10 min, followed by permeabilization with saponin-containing buffer. Plates were incubated overnight at 4 °C in 100 μL of permeabilization buffer containing a monoclonal anti-EBOV glycoprotein (clone 15H10, BEI resources, Manassas, VA, USA) at 1:3200 dilution followed by washing and a two-hour room-temperature incubation with secondary anti-mouse-HRP (Jackson ImmunoResearch, West Grove, PA, USA, #115-035-062) diluted 1:1000. Foci were scanned and quantitated on a Biospot plate reader (CTL, Shaker Heights, OH, USA).

### 2.11. RNA Isolation and qRT-PCR

RNA was isolated using TRIzol reagent from Invitrogen following the manufacturer’s instructions. RNA was subsequently converted to cDNA with the High-Capacity cDNA Reverse Transcription kit (#4368814). A total of 1 μg of RNA was used as input for each reaction. Quantitative PCR was performed using the PowerUp™ SYBR™ Green Master Mix (Applied Biosystems/Thermo Fisher Scientific, Waltham, MA, USA, #A25742) according to the manufacturer’s specifications and utilizing a QuantStudio™ 3 Real-time PCR machine from Applied Biosystems. 20 ng of cDNA were amplified. Duplicate qRT-PCR analyses were performed for each sample, and the obtained threshold cycle (CT) values were averaged. Gene expression was normalized to the expression of the housekeeping gene (Cyclophilin A, *CypA*) resulting in the ΔCT value. The relative mRNA or viral RNA was calculated by 2-ΔCT. The primers utilized in this study are as follows, 5′ to 3′ in format: *CypA*^for^: GCT GGA CCA AAC ACA AAC GG, *CypA*^rev^: ATG CTT GCC ATC CAG CCA TT, EBOV NP^for^: CAG TGC GCC ACT CAC GGA CA, EBOV NP^rev^: TGG TGT CAG CAT GCG AGG GC, *Clec4f*
^for^: ACA ACT CTG GAC ACG ACA ATC A, *Clec4f*^rev^: ATC TGT ACC TCC TTG TGA CAG C, *Timd4*^for^: GGG GAA GGT CCA GTT TGG TG, *Timd4*^rev^: TCC AAG CGC ACA TTC TTC TTG, *Clec2a*^for^: GCG GAA CCT GCC TCT TCT TG, *Clec2a*^rev^: GAT ACT TTT GCT GTG TGA CCG ACA T, *Irf1*^for^: GCC ATT CAC ACA GGC CGA TAC, *Irf1*^rev^: GCC CTT GTT CCT ACT CTG ATC C, *Gbp5*^for^: CCC AGG AAG AGG CTG ATA G, *Gbp5*^rev^: TCT ACG GTG GTG GTT CAT TT, *Gbp2a*^for^: CTG GCT CTG AGA AAA GGA ACT GA, *Gbp2a*^rev^: GAA AGT TGC TTC CTG TCT CCA, *Arg1*^for^: CAA ATT GTG AAG AAC CCA CGG, *Arg1*^rev^: CTT CCA ACT GCC AGA CTG TG, *Ym1*^for^: AGC TTT TGA GGA AGA ATC TGT GG, *Ym1*^rev^: CCT GAA TAT AGT CAA GAG ACT GAG A, *Clec10a*^for^: CCA AGA GCC TGG TAA AGC AGC, *Clec10a*^rev^: ATC CAA TCA CGG AGA CGA CC

### 2.12. Generation of and Studies Using IFN-γ Conditioned Media

ImKC-VP30 cells were plated at 50,000 cells/well in a 48-well format in RPMI with 5% FCS and pen/strep. The following day, some wells were treated with 20 ng/mL of recombinant murine IFN-γ for 24 h. IFN-γ-containing media were removed after 24 h, cells were washed once with media and maintained for another 24 h period in media. These media, called the conditioned media, were filtered through a 0.45 μm filter and either used directly or frozen at −80 °C until use. 

Prior to EBOV ΔVP30 infection, ImKC-VP30 cells in a 48-well format were held in media or treated for 24 h with 20 ng/mL of IFN-γ. Prior to infection, IFN-γ was removed and media refreshed. At the time of infection, additional wells of cells were treated with 20 ng/mL of IFN-γ or conditioned media. These cells were infected with EBOV ΔVP30 at the MOIs noted in the figures. EBOV ΔVP30-infected cells were assessed for GFP expression at 48 h following infection.

### 2.13. Statistical Analysis

Data analysis was performed using GraphPad Prism 9.4.1 (GraphPad, San Diego, CA, USA). Unless indicated otherwise, data are shown as mean ± SD. Unpaired one-tailed Student *t*-test was used to determine the statistical significance of single experiments. One-way ANOVA with Dunnett’s post hoc test was used to perform multiple comparisons against reference controls. Tukey’s post hoc test was used to perform multiple comparisons against every condition. In all tests, values of * *p* < 0.05, ** *p* < 0.01, *** *p* < 0.001, **** *p* < 0.0001 were considered significant.

## 3. Results

### 3.1. Characterization of Primary and Immortalized Kupffer Cells

To assess the expression of macrophage and KS-specific associated markers, we purified myeloid cells from other liver cells of a C57BL/6 mouse with a Percoll gradient. We phenotypically characterized these cells by flow cytometric analysis following the gating strategy in Appendix A. Approximately 50% of live cells were positive for CD45 (lymphocyte common antigen). Further analysis of the CD45^+^ cells indicated that ~30% of the cells were KCs, characterized by the expression of the KC-specific markers, CLEC4F (c-type lectin domain family 4 member F) and TIM-4 (T cell immunoglobulin and mucin domain containing 4) (Figure 1A and Appendix A) [26,27,28,29]. Analysis of the CLEC4^+^ or TIM-4^+^ cells demonstrated expression of macrophage markers F4/80, TLR4 and CD14 as well as the KC-associated marker CLEC2 (c-type lectin domain family 2, coded by *Clec1b*) (Figure 1A). 

Expression of these markers was also assessed in a murine Kupffer cell line (ImKC) that was established from transgenic mice expressing the thermolabile mutant tsA58 of simian virus 40 large T antigen (gating strategy on Appendix A). As others have previously shown [30], ImKCs were found to express F4/80, CD11B, TLR4 and CD14 (Figure 1B). Additionally, ImKCs expressed CLEC4F and a portion of the population expressed TIM-4 and CLEC2 (Figure 1B).

### 3.2. Primary KCs and ImKCs Respond to Polarizing Cytokine Treatments

Exposure of macrophages to certain cytokines drives macrophage polarization. In our earlier work, we demonstrated that interferon-γ (IFN-γ) generates a proinflammatory M1 phenotype in resident peritoneal macrophages as assessed by elevated production of interferon-stimulated genes (ISGs) such as interferon regulatory factor 1 (IRF-1) and other proinflammatory proteins [18]. In contrast, interleukin-4/interleukin-13 (IL-4/IL-13) treatment generates an M2a phenotype that is notable for arginase-1 (ARG-1) expression [17]. We evaluated the effect of these cytokines on primary KC polarization. Primary KCs were enriched from the bulk population obtained after the Percoll gradient by adherence on tissue culture plates for 2 h and mock-treated or cultured in the presence of IFN-γ or IL4/IL-13. Treatment of these cells with IFN-γ for 24 h stimulated *Irf-1* and guanylate binding protein 5 (*Gbp5*) transcript levels as anticipated (Figure 2A). The KCs that received a 24 h treatment of IL-4/IL-13 expressed Arginase 1 (*Arg-1)* and chitinase-like protein 3 (*Chil3*) (Figure 2A). These data indicate that these cells were appropriately responsive to the immunomodulatory cytokines.

Similarly, following 24 h IFN-γ and/or IL-4/IL-13 treatment, ImKCs polarized towards M1-like or M2-like phenotype, respectively (Figure 2B). Treatment with IFN-γ significantly elevated levels of *Irf-1* and *Gbp5* when compared to non-treated ImKCs. Moreover, significantly elevated levels of *Arg-1* transcripts were found in M2a-polarized ImKCs. Macrophage galactose-type lectin (*Clec10A*), another known M2a-associated marker [31], was significantly upregulated in M2a ImKCs when compared to M0 and M1 polarized ImKCs. Of note, basal (M0) levels of these activation markers were notably higher in the primary KCs compared to the immortalized cells. This may be due to a generalized activation of primary KCs that occurs during the isolation procedure. In total, these findings show that, similarly to primary macrophages, ImKCs can be polarized towards M1- and M2a-like cells. 

### 3.3. Primary and Immortalized Murine Kupffer Cells Support EBOV Infection

To examine if these cells are susceptible to authentic EBOV infection and the impact of IFN-γ on infection, primary KCs were pre-treated with IFN-γ. Cytokine-treated and untreated cells were infected with a multiplicity of infection (MOI) of 0.1 of EBOV-eGFP particles under maximum biocontainment for 48 h, GFP expression was observed in infected cells, and virus load was readily detected (Figure 3A,B). These findings demonstrate for the first time that cultured KCs support EBOV infection. Following IFN-γ treatment, 48 hpi EBOV virus loads in primary KCs trended lower, but the drop in viral load did not achieve statistical significance.

Similar experiments were performed with ImKCs. As evidenced in the micrographs, these cells were appreciably smaller than the primary KCs, but they also readily supported EBOV-eGFP infection (Figure 3C). In these cells, IFN-γ significantly diminished EBOV-GFP virus load by more than 10-fold (Figure 3C). Altogether, our data show that immortal and primary KCs are permissive for EBOV infection and that ImKC’s permissiveness to EBOV is significantly suppressed by prior IFN-γ treatment.

### 3.4. EBOV VP30-Expressing ImKCs Support EBOV ΔVP30 Infection

To establish a system to study EBOV infection using infectious viruses without requiring access to a maximum biocontainment laboratory, we utilized the biologically contained, previously developed model of EBOV infection referred to as EBOV ΔVP30 [24]. The biologically contained virions express a GFP reporter instead of the VP30 gene and the requisite VP30 gene is supplied in trans in the target cell. EBOV VP30-expressing ImKCs that were generated and biologically cloned were termed ImKCs-VP30. EBOV ΔVP30 stocks that were produced and titered in previously characterized Vero-VP30 cells [24] were evaluated in ImKC-VP30 cells. While a multiplicity of infection (MOI) of 1 as determined on Vero VP30 cells resulted in modest levels of GFP-positive cells at 60 h, a higher MOI of 10 resulted in ~40% of the cells being infected as assessed by flow cytometry (Figure 4A,B). Viral loads trended similarly as assessed by qRT-PCR (Figure 4C). To examine the ability of EBOV ΔVP30 to spread in ImKC-VP30 cells, the virus was added at several lower MOIs and monitored over time. Spread within the culture was observed, with increasing GFP intensity over time (Figure 4D). Production of new EBOV ΔVP30 was also assessed in the supernatant of these cells. Virus production was dependent on the quantity of input virus. By 3–4 days of infection, the quantity of the new virus in the supernatant plateaued at a modest level of ~10^3^ iu/mL (Figure 4E). Comparative studies of the infectious virus produced in supernatants on day 5 of infection demonstrated the importance of VP30-expression for the generation of EBOV ΔVP30 and the difference in the production of the virus in the ImKC-VP30 line versus the previously described Vero-VP30 line [22] (Figure 4F). Thus, while ImKC-VP30 cells support EBOV ΔVP30 infection that spreads through the culture, low levels of virion input resulted in modest generation of new infectious virions in supernatants.

### 3.5. IFN-γ Inhibits EBOV ΔVP30 Infection of ImKCs 

In a manner similar to authentic EBOV in the parental ImKCs, 24 h pre-treatment of ImKCs-VP30 with IFN-γ significantly downregulated viral loads of EBOV ΔVP30 as well as virus-driven GFP expression (Figure 5A–C and Appendix A). The duration of the antiviral effect of IFN-γ was evaluated in this infection system using confluent wells of ImKC-VP30 cells incubated with low-serum-containing media to reduce overgrowth of the culture. ImKCs-VP30 cells were treated for 24 h with IFN-γ (20 ng/mL). The cytokine was removed and fresh media were added. At 0–144 h following the completion of the IFN-γ treatment, cells were infected with EBOV ΔVP30 (MOI = 10) and infection was assessed by virus load and GFP expression at 60 h. When virus infection was initiated immediately after IFN-γ treatment, IFN-γ elicited a ~30-fold reduction in EBOV ΔVP30 virus load (Figure 5A). With time, the inhibitory effect of IFN-γ on EBOV ΔVP30 infection was reduced, with a ~3-fold reduction in virus load observed by 96 h after treatment. This ~3-fold inhibition persisted for at least 144 h (~6 days) following IFN-γ treatment. Similar, but more modest, trends were observed if virus infection was measured by the number of GFP^+^ cells in the infected cultures (Figure 5C). Cell viability was assessed over time and was not impacted by the length of the experiment or treatment with IFN-γ (Appendix A). These results indicate that much of the anti-EBOV activity elicited by IFN-γ is lost within 24 h; however, a more modest antiviral effect persists for as long as 6 days. A likely scenario to explain this is that some IFN-γ-elicited interferon-stimulated genes (ISGs) are only transiently expressed, whereas others continue to be expressed for a longer time.

To investigate if some common IFN-γ-elicited ISGs had prolonged expression, we measured *Irf1*, *Isg15*, guanylate binding protein 5 (*Gbp5*), and guanylate binding protein 2a (*Gbp2a*) transcripts levels over time. We previously demonstrated that IFN-γ stimulates the production of these transcripts in IFN-γ-treated murine peritoneal macrophages and overexpression of IRF1 and GPB5 inhibits EBOV infection [18]. In ImKC-VP30 cells, expression of all four ISGs was elevated by 24 h IFN-γ treatment when compared to untreated cells, ranging from a less than a 10-fold increase in *Isg15* expression to a more than 1000-fold increase in *Gbp5* and *Gbp2a* (Figure 5D–G, dotted line in each panel denotes baseline values found in untreated, uninfected cells for each ISG). Unexpectedly, the elevated levels of *Isg15* transcripts elicited by IFN-γ treatment did not change over six days, indicating prolonged expression of these transcripts. Transcripts of the transcription factor *Irf1* only modestly decreased (3-fold decrease) and were only statistically significant in infected cells at days 4 and 6 following IFN-γ treatment. With evidence of persistence of *Irf1* expression over the 6-day period and as *Irf1* is a transcription factor that drives expression of many IFN-γ-elicited ISGs [32], this suggests that the ISGs important for robust EBOV inhibition may be *Irf1*-independent. Levels of *Gbp2a* did decrease, with a ~9-fold drop by day 4, but transcript levels still remained orders of magnitude higher than levels found in the untreated cells. Expression of *Gbp5* also trended downward, but the decrease was not statistically significant.

### 3.6. Secreted ISGs Do Not Contribute to Protection against EBOV ΔVP30 Conferred by IFN-γ

It is appreciated that the expression of hundreds of ISGs is elicited upon IFN-γ treatment of macrophages [18]. Many of the proteins made from the ISGs remain cell-associated and are cytosolic, nuclear or membrane-associated. In contrast, some ISGs are secreted. A number of secreted ISG proteins are chemokines that do not have direct antiviral activity, but instead recruit adaptive immune cells to sites of infection. Other secreted proteins have direct antiviral activity which can be measured in tissue culture. To determine the role of secreted ISG proteins in the direct antiviral effect of IFN-γ, ImKC-VP30 cells were treated with IFN-γ for 24 h. Cytokine-containing media were removed, cells were washed, and fresh cytokine-free media were added back for 24 h. These conditioned media were collected and, in parallel with IFN-γ treatment, evaluated for their antiviral efficacy in ImKC-VP30 against EBOV ΔVP30 (Figure 6A). As anticipated, infection was inhibited by a 24 h pretreatment with IFN-γ. In contrast, conditioned media demonstrated no impact on levels of EBOV ΔVP30 infection as assessed by GFP^+^ cells at 48 h (Figure 6B). These findings indicate that secreted ISGs from ImKCs do not contribute to the antiviral effect conferred by type II IFN against EBOV ΔVP30.

In these studies, we also assessed the efficacy of a 24 h IFN-γ pretreatment compared to IFN-γ addition at the time of infection. We found that a 24 h pretreatment with IFN-γ was significantly more effective at inhibiting virus replication than the addition of IFN-γ at the time of infection, providing evidence that the ISGs elicited by IFN-γ pretreatment are responsible for controlling EBOV ΔVP30 infection (Figure 6B). 

## 4. Discussion

Here, we established an immortalized murine macrophage model to study innate immune responses during EBOV infection outside BSL4 facilities. We demonstrate that ImKCs express macrophage markers and treatment with IFN-γ or IL-4/IL-13 polarizing cytokines increased the expression of respective M1 and M2 markers on these cells, indicating that this cell line serves as an excellent macrophage model for studying the cytokine microenvironment. We further demonstrate that ImKCs and EBOV VP30-expressing ImKCs are permissive to EBOV-eGFP and EBOV ΔVP30, respectively, and show that IFN-γ treatment of these cells reduced viral loads and GFP viral gene expression. 

We have previously shown that IFN-γ treatment of primary mouse peritoneal macrophages robustly inhibits EBOV infection [18]. Here, in our ImKCs models, IFN-γ treatment reduced EBOV-eGFP and EBOV ΔVP30 in a similar fashion. We examine the duration of IFN-γ antiviral activity in our IFN-γ-treated ImKCs. Our data support the contention that there are at least two subsets of ISGs driving anti-EBOV activity. The first subset profoundly inhibited virus infection and is transient following IFN-γ treatment; within 24 to 48 h, this activity wanes. A second subset of ISGs had more prolonged, but less effective, inhibitory activity that persisted for the duration of our experiments. Analysis of four well-established IFN-γ-stimulated ISGs indicated that expression of these ISGs remained elevated over the 6-day experiment, with modest decreases in two of the transcripts, *gbp5* and *gbp2a*. However, the transcription factor, *Irf1*, that stimulates the expression of many known IFN-γ-dependent ISGs [32], remained elevated throughout the experiment, suggesting that the first wave of strong antiviral activity may be driven by ISGs that are not regulated by *Irf1*. 

In general, type I and II IFN responses are thought to be quite transient, yet transcripts of these four IFN-γ-elicited ISGs we examined were increased over basal levels in the ImKC-VP30 cells for as long as 6 days following treatment. Others have also reported prolonged IFN responses in other cell lines following either type I IFN treatment or virus infection [33,34,35]. Type I IFN treatment of Daudi cells was demonstrated to elicit long-term (7-day) expression of ISGs than that observed in several other cell types [34]. Studies in HUVECs have also demonstrated that the ISGs *MxA*, *Irf3,* and *Irf7* are robustly expressed for as long as 7 days during Hantaan virus infection, despite quite transient expression of both IFN-α and IFN-β [35]. Future studies to examine the duration of the antiviral activity in primary macrophages and identify the ISGs responsible for antiviral activity against EBOV are warranted. 

Our studies also demonstrate that secreted ISGs do not participate in the direct control of EBOV ΔVP30 infection, as conditioned media from IFN-γ-treated ImKC-VP30 cells conferred no protection against this virus. These studies provide insights into which ISGs are important for controlling EBOV ΔVP30 infection, implicating cell-associated ISGs in the protection conferred. Our studies also indicated that a 24 h IFN-γ pretreatment of ImKCs had more effective antiviral activity than adding IFN-γ at the time of infection. This finding suggests that IFN-γ-elicited ISGs present in ImKCs at the time of infection strongly contribute to the antiviral activity. 

Efforts in improving our understanding of mechanisms driving disease pathogenesis following EBOV infection have been hampered by the necessity of high biocontainment conditions (BSL4). The use of recombinant vesicular stomatitis virus (VSV) expressing the EBOV glycoprotein (rVSV/EBOV GP) has been useful for the study of glycoprotein-mediated processes such as viral entry and fusion, as well as adaptive immune responses towards EBOV GP [19,36,37]. However, the data obtained by using this infectious BSL2 model may not always recapitulate infection-mediated responses following authentic EBOV infection. Importantly, the generation of biologically contained EBOV lacking the VP30 gene (EBOV ΔVP30) that recapitulates EBOV morphology and growth properties permits EBOV studies in tissue culture under lower containment conditions [24]. However, a small animal model (e.g., mouse) suitable for work with EBOVΔVP30 that could be employed outside of the BSL4 is still needed. 

In summary, we show that KCs, ImKCs and EBOV ΔVP30-expressing ImKCs support infection with EBOV and EBOV ΔVP30, respectively. Furthermore, our in vitro studies demonstrated that IFN-*γ* inhibits EBOV and EBOV ΔVP30 infection in ImKCs and provides insights into the type of ISGs that are responsible for their antiviral activity. Overall, these studies provide new tools for the study of EBOV infection that can potentially aid the development of anti-filovirus therapeutics. 

## Figures and Tables

**Figure 1 viruses-15-02077-f001:**
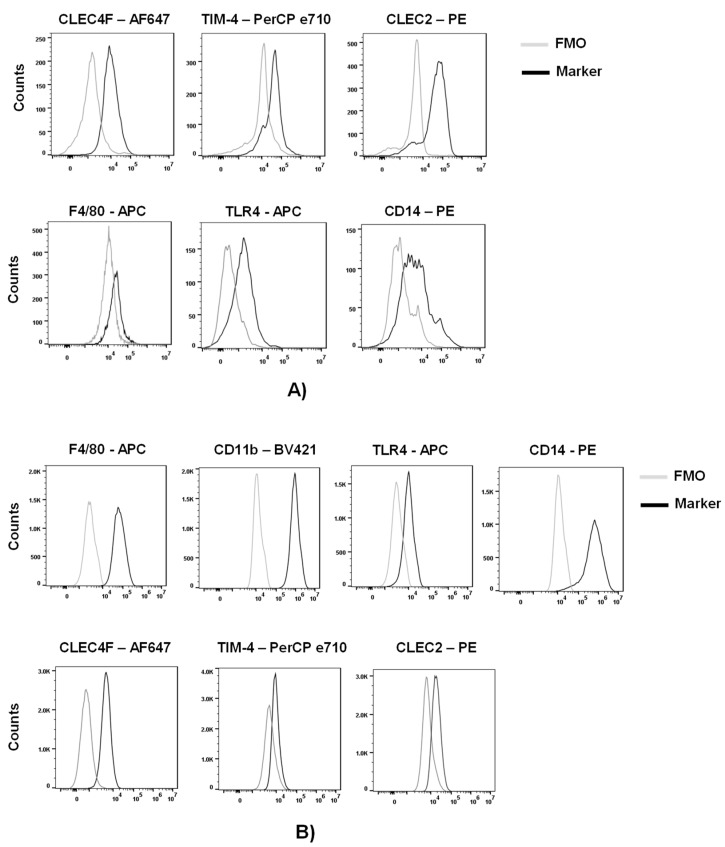
Phenotypic characterization of primary and immortal Kupffer cells. (**A**) Myeloid cell populations were isolated from livers of C57BL/6 mice and analyzed by flow cytometry following the gating strategy in Appendix A. Live CD45^+^ CLEC4F^+^ or Live CD45^+^ TIM-4^+^ primary KC cells were analyzed for the expression of CLEC4F, TIM-4, F4/80, TLR4, CD14 and CLEC2 by flow cytometry (*n* = 3; two-livers pooled per group). (**B**) ImKCs were phenotypically characterized by flow cytometry for the expression of macrophage and Kupffer cell-specific markers F4/80, CD11b, TLR4, CD14, CLEC4F, TIM-4 and CLEC2 following the gating strategy depicted in Appendix A. Shown are representative flow cytometry plots from three independent biological experiments. Fluorescent minus one (FMO) controls were used to delineate gates and served as negative controls for their respective marker expression comparison.

**Figure 2 viruses-15-02077-f002:**
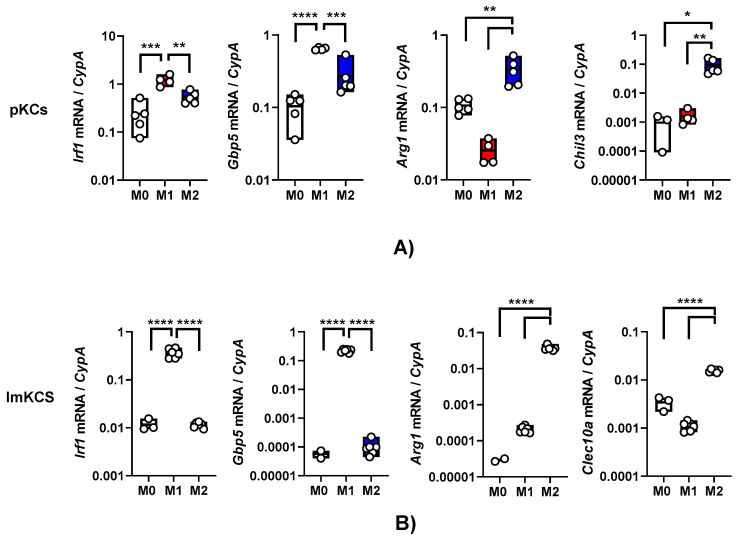
Primary and immortal Kupffer cells polarize towards M1- and M2a- like cells following treatment with appropriate cytokines. Gene expression changes in (**A**) primary KCs or (**B**) ImKCs following 20 ng/mL IFN-γ (M1) or 20 ng/mL of IL-4/IL-13 (M2a) treatment of KCs. Gene expression levels of *Irf1*, *Gbp5*, *Arg1*, *Chil3* and *Clec10a* were determined by qRT-PCR. Cyclophilin A (*CypA*) was used as a reference gene. *p*-values were determined by unpaired one-tailed *t*-test comparing individual treatments with unstimulated (M0) controls. (* *p* < 0.05; ** *p* < 0.01; *** *p* < 0.001; **** *p* < 0.0001).

**Figure 3 viruses-15-02077-f003:**
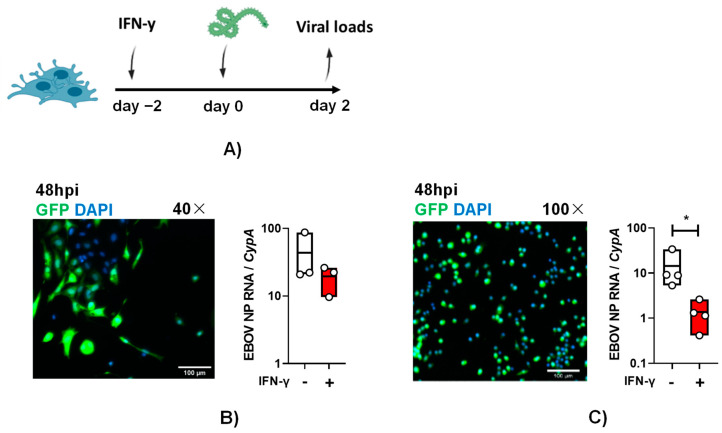
Primary and immortal Kupffer cells support EBOV infection and IFN-γ pretreatment decreases virus infection of ImKCs. (**A**) Schematic of study. Primary KCs (**B**) or ImKCs (**C**) were left untreated or treated with IFN-γ (20 ng/mL) for 48 h and then infected with 1 × 10^4^ EBOV-eGFP particles (*n* = 3, three independent biological experiments conducted at maximum containment laboratory). Shown in panels (**B**,**C**) are virus-driven GFP expression (micrographs) and viral loads after 48 hpi. Gene expression levels of EBOV NP were determined by qRT-PCR of cell lysates with cyclophilin A used as a reference gene. *p*-values were obtained by unpaired one-tailed *t*-test. (* *p* < 0.05). (**A**) Created with BioRender.com.

**Figure 4 viruses-15-02077-f004:**
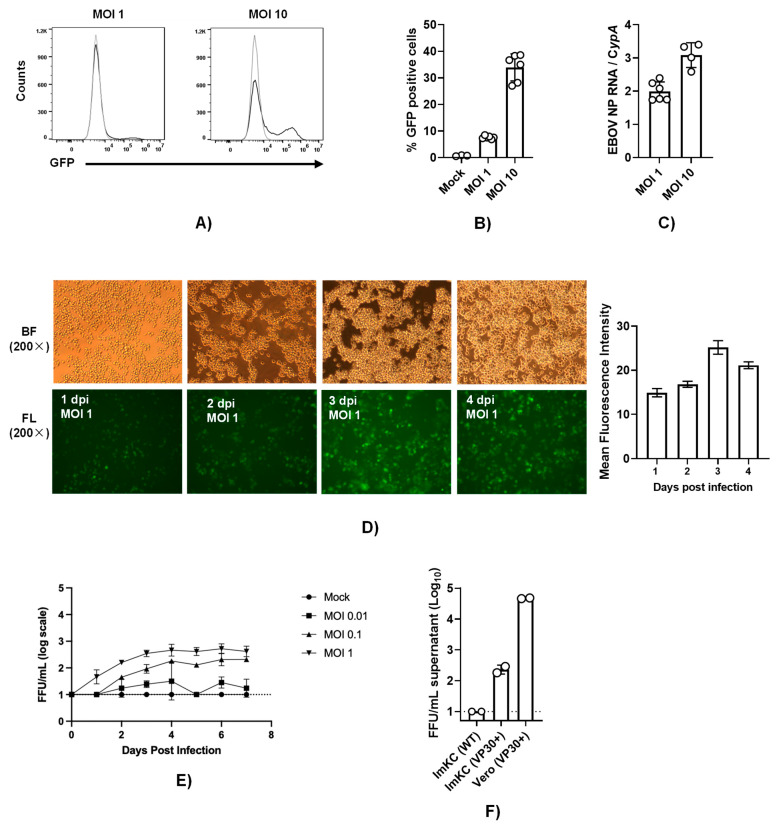
ImKCs-VP30 are susceptible and permissive to EBOV ΔVP30 infection. (**A**,**B**) ImKCs-VP30 cells were infected with an MOI of 1 or 10 with EBOV ΔVP30 for 60h (*n* = 3, three independent biological experiments). (**A**,**B**) GFP expression was determined by flow cytometry. (**C**) Gene expression levels of EBOV NP were determined by qRT-PCR. Cyclophilin A was used as a reference gene. Data represents the mean ± SD. (**D**) GFP expression over the course of a 4-day infection. Shown are 200× micrographs of white (top panel), fluorescent (bottom panel) light images and quantification of mean fluorescence intensity by ImageJ v1.54d (right panel). (**E**) Infectious virus present in supernatants collected over time beginning with an MOI of 0.01 to 1. Supernatants were titered on Vero-VP30 cells. (**F**) A comparison of infectious virus produced in supernatants on day 5 from ImKCs, ImKC-VP30s and Vero-VP30 cells infected with an MOI of 0.1 of EBOV ΔVP30.

**Figure 5 viruses-15-02077-f005:**
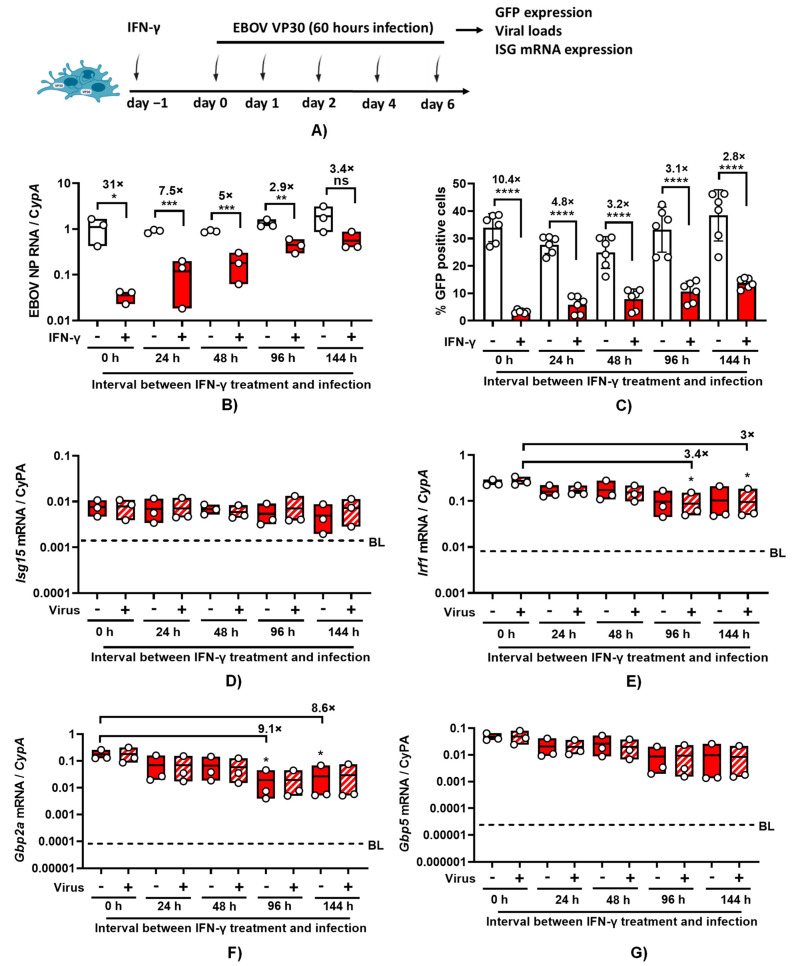
EBOV ΔVP30 infection in ImKC-VP30 cells is inhibited by IFN-γ and the effect is gradually reduced over time. EBOV VP30-expressing ImKCs were polarized with IFN-γ (20 ng/mL) for 24 h, media replaced and EBOV ΔVP30 infection and expression of four ISGs were analyzed at the different times noted. (**A**) Schematic of the experiment. (**B**) Expression of EBOV NP following infection with EBOV ΔVP30 (MOI = 10) for 60 h (*n* = 3, three independent biological experiments). Cyclophilin A was used as a reference gene. (**C**) EBOV ΔVP30 infection measured by eGFP expression. Data shown represent the mean ± SD. (**D**–**G**) Expression of *Isg15*, *Irf1*, *Gbp2a* and *Gbp5* were determined by qRT-PCR in EBOV ΔVP30-infected and uninfected cells following IFN-γ treatment. Cyclophilin A was used as a reference gene. Dotted line labeled with BL in each panel denotes baseline levels of expression found in untreated, uninfected cells for each ISG. (**B**,**C**) *p*-values were obtained by unpaired one-tailed *t*-test. (**D**–**G**) *p*-values were obtained by one-way ANOVA, Dunnett’s post hoc test. (* *p* < 0.05; ** *p* < 0.01; *** *p* < 0.001; **** *p* < 0.0001; ns = no significance). (**A**) Created with BioRender.com.

**Figure 6 viruses-15-02077-f006:**
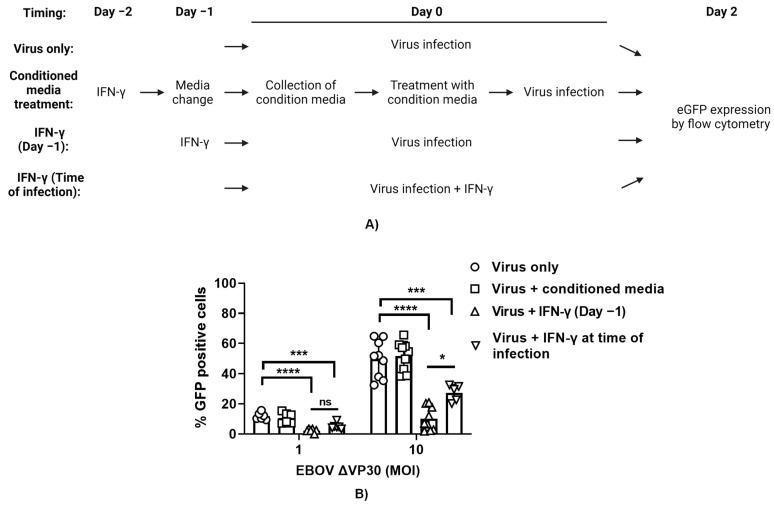
IFN-γ-induced ISG secretome does not contribute to protection conferred by IFN-γ against EBOV ΔVP30. (**A**) Schematic of the experiment. (**B**) ImKCs were untreated or treated with IFN-γ 24 h prior to infection, IFN-γ at the time of infection or conditioned media from ImKCs treated with IFN-γ. Cells were infected with EBOV ΔVP30 (MOI 1 or 10) for 48 h. Data represent the mean ± SD. *p*-values were obtained by one-way ANOVA (Dunnett post hoc test, *** *p* < 0.001; **** *p* < 0.0001) (Tukey post hoc test, * *p* < 0.05; ns = no significance). (**A**) Created with BioRender.com.

## Data Availability

Data is contained within the article or supplementary material.

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
