# Peer review of "Effect of Interferon Gamma on Ebola Virus Infection of Primary Kupffer Cells and a Kupffer Cell Line"

_viruses, 2023, doi:10.3390/v15102077_

Round 1

Reviewer 1 Report

José A. Aguilar-Briseño et al. investigated EBOV infection in mouse liver resident macrophages and proposed new mouse macrophage cell lines suitable for experiments in low containment facilities. Using these in vitro systems, they further investigated the antiviral effects of IFN-γ. Their manuscript is overall well written and presents relevant data. However, the following points need to be improved or clarified.

TITLE

The title should be modified since “Ebola virus infection of liver cells” is too broad since EBOV also infects hepatocytes and the current title does not represent the data described in this study. For example, “Effect of Interferon gamma 2 on Ebola virus infection of primary Kupffer cells and a Kupffer cell line” may be better.

ABSTRACT

L19. “EVD disease”: The abbreviation “EVD” already contains “disease”. This need to be corrected.

INTRODUCTION

L41-42. “...the Orthoebolavirus comprised one genus composed of six viral species…”: This sentence is hard to understand; please rephrase it. Orthoebolavirus is the genus. In addition, EBOV is the abbreviation of Ebola virus and cannot be used to abbreviate the species name “Orthoebolavirus zairense. One possible modification is: Orthoebolavirus zairense, Orthoebolavirus Sudanense, Orthoebolavirus Bundibugyoense, Orthoebolavirus Taiense, Orthoebolavirus Restonense and Orthoebolavirus Bombaliense (1–3). Of these, Ebola virus (EBOV) representing the species Orthoebolavirus zairense has the greatest…

L52. “…confers >95%...”: This should be “confers >95% protection”.

MATERIALS AND METHODS

L114. “was mashes” should be “was mashed”.

L118. What was the composition of the laboratory-made lysis buffer? This should be indicated.

L120. “cells were plated and send….” should be “cells were plated and sent…”.

L121-122. Please rephrase the sentence to improve the readability.

L130. More details should be provided for the plasmid, VP30-encoding pLV-EF1α-VP30-IRES-hygro_v2.1-hygro-130 mycin. How was it constructed?

L152. “All” should be “all”.

L197. “(1x104 cells)” should be corrected.

L240. “IFN-γ containing media” should be “IFN-γ-containing media”.

L242. “0.45 μ” should be “0.45 μm”.

L253. Are the authors referring to “median”? But, it does not seem that the median is used in any of the figures. This should be clarified.

RESULTS

L330. “Similar studies” should be “Similar experiments”.

L344. This MOI may be based on the titer determined on Vero E6. So, it is not actually MOI =1 on other cells.

L406-407. “expression of all four ISGs was dramatically elevated by 24-hour IFN-γ treatment when compared to untreated cells”: In Figure 5, only baseline levels obtained from mock infected cells are shown. The data shows higher levels than control but does not show “elevated” level unless the levels before IFN-γ treatment are shown. This sentence should be rephrased.

L444-449. There is no data to support this experiment or conclusions arising from it. Also, materials used in this experiment have not been mentioned in the methods.

DISCUSSION

L500. “ImKC-vp30” should be “ImKC-VP30”.

L520. “Efforts in improve” should be “Efforts in improving”.

L522-524. “The used of recombinant vesicular stomatitis……..has been useful for the study of glycoprotein-mediated process….” Please rephrase this sentence.

L533. “EBOV ΔVP30 expressing ImKC” should be “EBOV VP30-expressing ImKC”.

There are many grammatical and typographical errors.

Reviewer 2 Report

This manuscript proposed and characterized a novel tool to study EBOV infection by utilizing Kupffer cells or immortalized mouse Kupffer cells. The findings comparing the results of infection from different types of viruses (authentic virus, delta-VP30 system, and the VSV pseudo-typed system) are also helpful for researchers in this field.

Minor suggestions:

Please proofread the manuscript thoroughly since many minor errors have been found. Examples are below.

1.      Line 52, >95% protection?

2.      Other citations: PMID: 34715022, PMID: 35303429

3.      Line 119, delete “for”?

4.      Line 152, please edit

5.      Line 171, please list the four viruses with strain information.

6.      Line 336, section 3.5 follows 3.3, please edit.

7.      Line 346, remove the period before the figure callout

8.      Line 349, stated here that most cells were infected by day 4. However, it is hard to tell by the image, which looks like only some cells were infected. Can you show a quantification of the infection percentage?

9.      Line 385, remove the period at the end.

minor edits needed
